**Data Availability Statement:** All the relevant data are within the manuscript and in the supporting files. Sequence analysis scripts and data are

# *Leptospira* enrichment culture followed by ONT metagenomic sequencing allows better detection of *Leptospira* presence and diversity in water and soil samples

Myranda Gorman[1], Ruijie Xu[2,3], Dhani Prakoso[1¤], Liliana C. M. Salvador[2,3,4], Sreekumari Rajeev🆔[1]*

1 Department of Biomedical and Diagnostic Sciences, College of Veterinary Medicine, University of Tennessee, Knoxville, Tennessee, United States of America, 2 Institute of Bioinformatics, University of Georgia, Athens, Georgia, United States of America, 3 Center for the Ecology of Infectious Diseases, University of Georgia, Athens, Georgia, United States of America, 4 Department of Infectious Diseases, University of Georgia, Athens, Georgia, United States of America

¤ Current address: Professor Nidom Foundation, Surabaya, Indonesia
* srajeev@utk.edu

## Abstract

### Background

Leptospirosis, a life-threatening disease in humans and animals, is one of the most wide-spread global zoonosis. Contaminated soil and water are the major transmission sources in humans and animals. Clusters of disease outbreaks are common during rainy seasons.

### Methodology/Principal findings

In this study, to detect the presence of *Leptospira*, we applied PCR, direct metagenomic sequencing, and enrichment culture followed by PCR and metagenomic sequencing on water and soil samples. Direct sequencing and enrichment cultures followed by PCR or sequencing effectively detected pathogenic and nonpathogenic *Leptospira* compared to direct PCR and 16S amplification-based metagenomic sequencing in soil or water samples. Among multiple culture media evaluated, Ellinghausen-McCullough-Johnson-Harris (EMJH) media containing antimicrobial agents was superior in recovering and detecting *Leptospira* from the environmental samples. Our results show that enrichment culture followed by PCR can be used to confirm the presence of pathogenic *Leptospira* in environmental samples. Additionally, metagenomic sequencing on enrichment cultures effectively detects the abundance and diversity of *Leptospira spp.* from environmental samples.

### Conclusions/Significance

The selection of methodology is critical when testing environmental samples for the presence of *Leptospira*. Selective enrichment culture improves *Leptospira* detection efficacy by PCR or metagenomic sequencing and can be used successfully to understand the presence and diversity of pathogenic *Leptospira* during environmental surveillance.

available at https://github.com/rx32940/Environmental_Lepto_detection. All the sequences are deposited in NCBI under BioProject, PRJNA868285, PRJNA868284 and PRJNA868283.

**Funding:** M.G. was supported by a scholarship from the Boehringer Ingelheim Veterinary Summer Scholar Program. Funding for this research was through the University of Tennessee, College of Veterinary Medicine startup funds for S.R. and the Office of Research and the University of Georgia for L.C.M.S. The funders did not have any roles in the study design, data collection and analysis, decision to publish, or preparation of the manuscript.

**Competing interests:** The authors have declared that no competing interests exist.

## Author summary

Leptospirosis, a life-threatening disease in humans and animals, is one of the most widespread global zoonosis. Contaminated soil and water are major sources of transmission in humans and animals. For this reason, clusters of disease outbreaks are common during the rainy season. In this study, *Leptospira* enrichment cultures followed by PCR and sequencing detected pathogenic and nonpathogenic *Leptospira* in soil and water samples. However, pathogenic and intermediate groups of *Leptospira* were more prevalent in soil samples tested. Metagenomic sequencing on enrichment culture was effective in detecting the abundance and diversity of various *Leptospira spp*. in environmental samples. Soil samples in proximity to water may be an ideal niche for *Leptospira* growth and survival and may be an appropriate sample of choice for testing.

## Introduction

Many species of *Leptospira*, a spirochete bacterium that causes leptospirosis, are maintained in the renal tubules of numerous mammalian species and the environment [1]. Leptospirosis is a life-threatening illness in humans, causing approximately 1 million cases and 60,000 deaths annually [2]. A variety of mammals following *Leptospira* infection may become clinically ill or remain as asymptomatic renal reservoirs of infection. They shed bacteria through urine and act as a source of infection to other animal hosts and environmental contamination [3]. The environmental route is the most common mode of *Leptospira* transmission in humans. Leptospirosis is endemic to tropical countries, and outbreaks occur during natural disasters when humans come into contact with the contaminated environment. The host and the environment interface may play a major role in the epidemiology and transmission of *Leptospira* infection. Continuous changes in climatic landscapes may increase the number of outbreaks occurring globally. The critical gap in knowledge on environmental persistence and cycling of *Leptospira* needs to be addressed [4]. A number of studies have been conducted to investigate the level and type of *Leptospira* commonly found in environmental samples by applying multiple techniques. The sensitivity and specificity of *Leptospira* detection in environmental samples can be complicated by the coexistence of chemical, physical, and biological contaminants. Low levels of *Leptospira* present in the environmental sample among abundant contaminant microorganisms can also lead to false-negative results. Therefore, improvements in methods are needed for the accurate detection of *Leptospira* in environmental samples. Recently with the advent of Next Generation Sequencing (NGS) methods, the assessment of the microbial composition of environmental samples for disease surveillance has become a routine practice. For example, Oxford Nanopore Technologies (ONT) has been widely used for disease and environmental surveillance [5–8]. We propose that combining traditional selective culture methods with advanced sequencing could improve *Leptospira* detection in the environmental samples. In this study, we evaluated multiple methods including selective enrichment culture, direct PCR, 16S rRNA gene amplification-based sequencing, direct metagenomic sequencing, and *Leptospira* enrichment culture followed by PCR and metagenomic sequencing to detect the presence of *Leptospira* DNA in environmental samples.

## Materials and methods

### Sample collection and processing

We collected representative soil and water samples in sterile containers from a local creek where abundant human and animal activity was observed. We collected soil from the damp

edge of the creek where water was collected and transported the samples to the laboratory on ice. After mixing one liter of the water thoroughly, we added 10 mL of Ellinghausen-McCullough-Johnson-Harris (EMJH) liquid medium (Becton Dickinson, Sparks, MD, USA) supplemented with Difco *Leptospira* Enrichment EMJH (Becton Dickinson, Sparks, MD, USA) to the top of the water sample and allowed it to incubate for three hours. This was done assuming that, *Leptospira* present in the water might be attracted to the media rich top of the sample. Then, 200 mL of water from the surface was collected and filtered with a large pore size (40 μm) nylon filter to remove larger debris. The flow through was again filtered using a 0.45 μm filter to remove other contaminant bacteria assuming that *Leptospira* with smaller size will be allowed to pass through. The filtrate was then divided into two 100 mL aliquots. We spiked one of the 100 mL aliquots with *Leptospira interrogans* serovar Copenhageni (final concentration of $10^7$ bacteria per mL) to use as the control, and the second aliquot was designated as the test sample. Direct PCR, culture, and sequencing was performed on these samples.

For the processing of soil samples, 25 g of soil was divided between two flasks and then mixed with 100 mL of phosphate-buffered saline (PBS). After mixing thoroughly for five minutes, the sample was allowed to settle for thirty minutes. Then 10 mL of EMJH media was added to the top of the samples and allowed to settle overnight. A longer settling time was required to obtain a cleaner sample for inoculation. Once settled, 80 mL of supernatant from each flask was collected and filtered through a 40 μm nylon filter followed by a 0.45 μm filter. The filtrate was then aliquoted into two 40 mL samples. We spiked one of the aliquots with *Leptospira interrogans* serovar Copenhageni (final concentration of $10^7$ bacteria per-mL) and designated it as "control". The non-spiked sample is designated as "test" sample. The aliquots from these samples were further used for direct PCR, culture, and sequencing.

### *Leptospira* detection using direct PCR from water and soil

The aliquots of test (non-spiked) and control (spiked) samples described above were centrifuged at 4,000 x g for forty minutes. The pellet was collected and then reconstituted with 10 mL PBS. Then 1 mL aliquots were pipetted into ten 1.5 mL collection tubes and stored at -20˚C. DNA was extracted from three replicates of the spiked and test samples using the Quick-DNA Fecal/Soil Microbe Miniprep Kit (Zymo Research, Irvine, CA, USA) following the manufacturer's protocol. The extracted DNA samples were then tested with Real-Time PCR targeting genes *lipL32*, and *16s rRNA* to confirm the presence of *Leptospira* DNA [9,10] using a Q qPCR Instrument (Quantabio, Beverly, MA, USA). The cutoff for a positive sample was set at a Cq value of 40. The DNA extraction and PCR were performed on two technical replicates.

### 16S rRNA gene-based metagenomic sequencing

We used a protocol based on a recent publication describing monitoring of fresh water for pathogens for this procedure [11]. Briefly, extracted DNA samples were amplified using the full length of *16s rRNA* gene primers with common primer binding sequences 27f and 1492r, attached to unique 24 bp barcodes and nanopore motor protein tether sequence. PCR was performed with 600 nM of each forward and reverse primer, 25 μL of Premix Taq DNA Polymerase (TakaraBio, Shiga, Japan), and a 10 μL DNA template in a 50 μL reaction. The amplification cycles used the following conditions: 94˚C for 2 minutes, followed by 35 cycles of 94˚C for 30 seconds, 60˚C for 30 seconds, and 72˚C for 45 seconds with final elongation at 72˚C for 5 minutes. The amplicons from the PCR step were purified using NucleoSpin Gel and PCR Clean-up (Macherey Nagel, Duren, Germany) following the manufacturer's protocol.

The barcoded amplicon samples were pooled in equimolar ratios, and library preparation and sequencing were conducted using Ligation Sequencing Kit SQK-LSK-109 (Oxford Nanopore Technologies, Oxford, UK) on the MinION (Oxford Nanopore Technologies, Oxford, UK) sequencing platform following the manufacturer's instructions.

## Metagenomic sequencing directly from the environmental samples

The samples were spun down at 4,000 x g for forty minutes, and the supernatant was discarded, leaving a pellet in 10 mL of supernatant. After thorough mixing, 1 mL was aliquoted into ten 1.5 mL microcentrifuge tubes then centrifuged at 14,000 x g for three minutes. The supernatant was removed from each tube, leaving 200 μL with the pellet. DNA was extracted using Quick-DNA Fecal/Soil Microbe Miniprep Kit (Zymo Research, Irvine, CA, USA) according to the manufacturer's instructions. The extracted DNA from water and soil underwent a further purification step using Monarch PCR & DNA Cleanup Kit (New England Biolabs, Ipswich, MA, USA). DNA Library preparation was conducted using Native barcoding genomic DNA Kit SQK-LSK 109 combined with EXP-NBD104 (Oxford Nanopore Technologies, Oxford, UK) following the manufacturer's instructions. After the end repair step, DNA from samples was barcoded and pooled in equimolar amounts to make one library, followed by adapter ligation and sequencing for approximately 48 hours on the MinION sequencing platform.

## *Leptospira enrichment* culture, followed by PCR and metagenomic sequencing

We tested multiple media and antimicrobial combinations to enrich and grow *Leptospira* from environmental samples. We used four commonly used *Leptospira* culture media: Stuart, Korthof, Fletcher, and Ellinghausen–McCullough–Johnson–Harris (EMJH) media. We tested each of these media with and without the addition of antimicrobials, 5-Fluorouracil (5-FU), and an antimicrobial cocktail (STAFF) to control the growth of competing bacteria in the cultures [12]. We inoculated 500 μL of the processed water and soil to each of these media. The cultures were then incubated in a 29˚C incubator for four weeks, monitored at 24 hours, 72 hours, and then once a week for four weeks using dark field microscopy (DFM). The samples with the presence of organisms exhibiting *Leptospira*-like motility and morphology were presumptively identified as positive for *Leptospira* and scored using a 0 to +4 rating system based on the number of spirochetes present (Table 1).

The presence and level of other contaminating bacteria were also recorded at each time point of evaluation. After four weeks of incubation and monitoring, 1 mL from each culture presumptively identified to contain *Leptospira*-like bacteria were collected, and DNA was extracted (Zymo Quick-DNA Miniprep kit, Zymo Research, Irvine, CA, USA). The DNA was then tested by PCR using *lipL32* and *16s rRNA* gene primers as described above.

**Table 1. The scoring system used in this study to evaluate cultures.**

| Scoring | The relative number of *Leptospira*-like organisms under DFM |
| --- | --- |
| 0 | None seen |
| +1 | Less than 25 |
| +2 | Between 25 and 50 |
| +3 | Between 50 and 100 |
| +4 | More than 100/too numerous to count |

To evaluate the composition and *Leptospira* diversity of the culture samples, we pursued metagenomic sequencing using DNA extracted from culture samples. A composite of positive samples of culture from water and soil was used to reduce the cost of testing. Briefly, extracted DNA was purified using SparQ PureMag Beads (Quantabio, Beverly, MA, USA) following the instruction from the manufacturer. The Native barcoding, genomic DNA Kit SQK-LSK 109 combined with EXP-NBD104 was used for library preparation. The samples underwent end-repair, barcode ligation for multiplexing, and adapter ligation and sequencing. The DNA sequencing was conducted using the MinION sequencing platform for approximately 24 hours. All the sequences are deposited in NCBI under BioProject, PRJNA868285 (Direct Meta-genomics Sequencing Environmental Water and Soil Samples), PRJNA868284 (16SrRNA sequencing in water and soil samples), and PRJNA868283 (ONT sequenced environmental water/soil Samples after culture enrichment).

## Sequence analysis

All scripts used for sequence data analysis are available at: https://github.com/rx32940/Environmental_Lepto_detection. All samples were base called using Guppy v. 6.1.1 with High Accuracy setting (https://community.nanoporetech.com). Samples were demultiplexed using Porechop v. 0.2.4. (https://github.com/rrwick/Porechop). To trim customized barcodes and adapters from each read during demultiplexing, customized barcodes and adapters were added to Porechop's Adapter.py file before demultiplexing. The command '—discard_middle" was specified to remove chimeric reads attached by two different barcodes. The quality of the filtered reads was assessed using NanoStat v 1.5.0 [13] and visualized using Pistis v 0.3.4 (https://github.com/mbhall88/pistis).

## Microbial composition profiling and *Leptospira* classification from 16S dataset

Since the length of bacterial 16S rRNA is around 1.5 kbp, reads smaller than 1.4 kbp and larger than 1.6 kbp were filtered using NanoFilt v. 2.8.0 [13] to remove potential existing contaminations. To classify each read's microbial taxon, each sample's filtered reads were mapped against SILVA v. 138.1 *16S rRNA* database [14] using Minimap2 v 2.17 [15] with the recommended option for Nanopore reads "-ax map-ont". Statistics for the percentage of reads mapped to the database were assessed using the "stat" function in Samtools v.1.10 [16]. Mapped Bam files were converted to Bed format using "sam2bed" function in BEDOPS v 2.4.39 [17] for the downstream analysis. Microbial composition and abundance for each sample were analyzed using R. Reads mapped to more than one microbial taxon were assigned to the lowest common ancestor (LCA) of all mapped taxa. Reads that could not be assigned to at least a family-level taxon were removed from the downstream analysis due to low discrimination. Reads classified under all the taxa belong to the same bacterial family were summarized to obtain each sample's microbial composition at the family level. The microbial composition for each sample was summarized and visualized using "dplyr" (https://github.com/tidyverse/dplyr) and "ggplot2" packages in R (https://ggplot2.tidyverse.org).

All reads mapped under phylum *Spirochaete* were extracted from each sample's sequences file using the "subseq" function in SEQTK v. 1.2 (https://github.com/lh3/seqtk) using read IDs. Extracted *Spirochaetota* reads were aligned with all *Leptospira 16S rRNA* sequences deposited in NCBI using MUSCLE v 3.8.0 [18]. Resulting multi-sequence alignment were used to build a maximum likelihood (ML) phylogeny using the IQ-Tree v. 1.6.12 with options "-bb 1000 -m MFP" for evolutionary model searching and bootstrapping support. ML phylogenies were visualized using the "ggtree" package in R [19].

### Microbial composition profiling and *Leptospira* classification and identification from direct Sequencing and Sequencing from the enrichment culture

Each sample's microbial composition was profiled using Kraken2 v. 2.0.9 [20] with the maxikraken2 database (https://lomanlab.github.io/mockcommunity/mc_databases.html) using the default settings. Profiling results of all samples were combined into a single file using KrakenTools v 1.2 (https://github.com/jenniferlu717/KrakenTools). Microbial reads classified under each taxon were analyzed and summarized in R using "dplyr" package (https://github.com/tidyverse/dplyr) and visualized using "ggplot2" package (https://ggplot2.tidyverse.org). All reads mapped under *Leptospira* taxa were subset from microbial profiles of each sample to visualize the relative percentage of *Leptospira* species identified from each sample.

## Results

### Direct PCR results from environmental samples

For the direct real-time PCR, both the test and spiked (control) water and soil samples were tested using the *Leptospira* specific *16s rRNA* and *lipL32* gene markers. The *16s rRNA* gene was amplified from all the samples, however, Cq values were high in the test samples suggesting low levels of *16s rRNA* (Fig 1). *LipL32* gene amplification was detected only in the control samples and not in the test samples.

### Culture results

The soil and water samples were cultured in four different types of media with microbial inhibitor combinations. The presence and levels of organisms with morphology compatible with

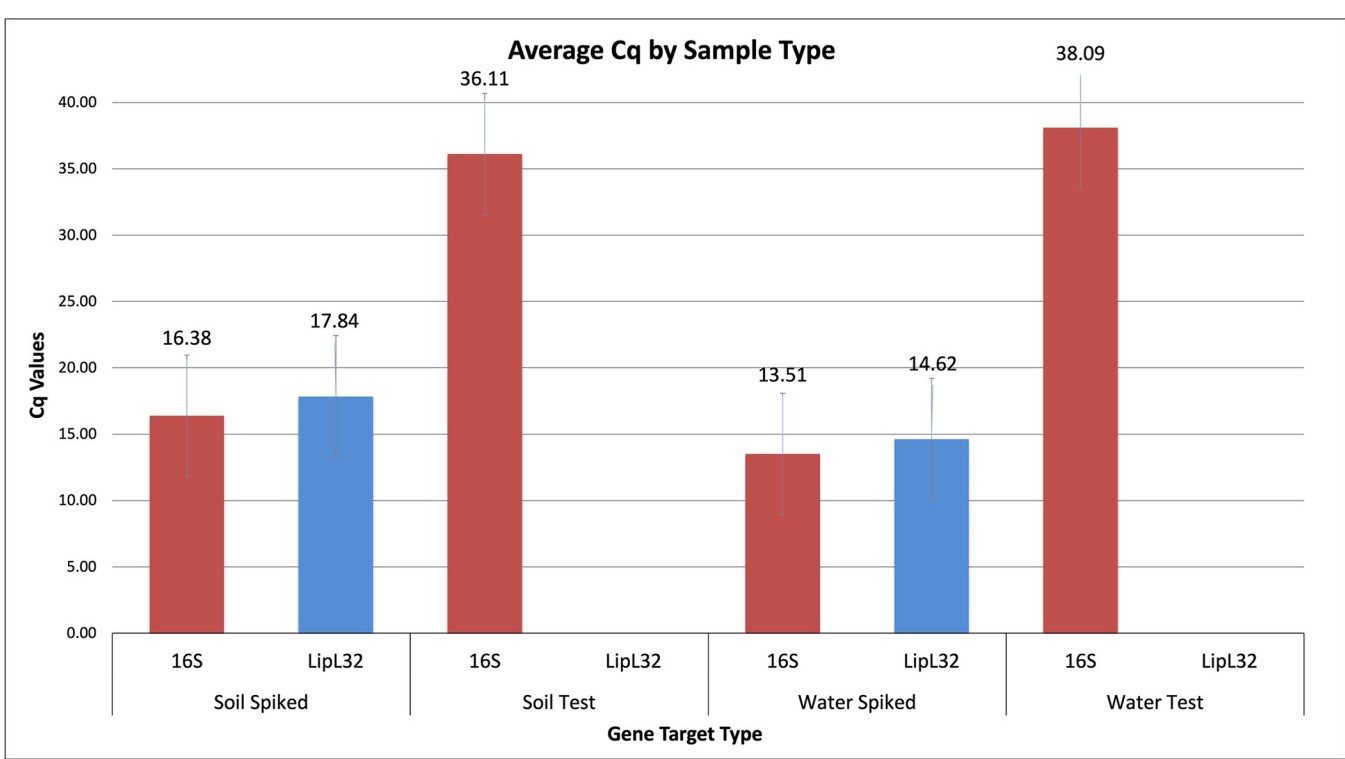

**Fig 1. Average Cq values for the water and soil samples that were directly tested with real-time PCR. Spiked soil and water samples were used as the control for testing and are the only samples where *lipL32* genes were detected. All samples displayed the presence of *16s rRNA* gene.**

*Leptospira* were recorded with a 0 to +4 ordinal system (Table 1). The cultures with selective antimicrobial inhibitors demonstrated large and earlier increases in bacterial organisms with morphology and motility compatible with *Leptospira* when observed under the DFM. Overall, EMJH cultures with 5-FU or STAFF were favorable for *Leptospira* growth for the water test group. For the soil test samples, the culture results were more variable. The usage of selective antimicrobials in the cultures did not have as much of a visible impact on the growth of *Leptospira* in the soil samples. Overall, Fletcher and EMJH media demonstrated favorable growth for the soil samples, with EMJH performing marginally better than the Fletcher media. All water test and soil test cultures were tested using real-time PCR to confirm the presence of *Leptospira*. The cultured test water and soil samples were positive for *lipL32* and *16s rRNA* gene markers. The Cq values for the samples were consistently around 30 to 35. The growth pattern of *Leptospira*-like organisms in various cultures is shown in S1 and PCR results are shown in S2 Files.

## Sequencing results

The details of results from all sequencing methods are shown in Table 2

## Microbial composition profiling and *Leptospira* classification from 16S dataset

A very low number of reads were classified under the phylum taxon "Spirochaetota" in the soil (1 read) and water (9 reads) samples when 16S rRNA gene sequence dataset was analyzed. The single reads identified in the soil sample were closely clustered with the reference *16S rRNA* sequences of two pathogenic (P1) *Leptospira spp.*, *L. interrogans* and *L. kirschneri* (bootstrap (bt) support: 99%). For the 9 reads obtained from the water samples, reads were clustered into two separate clusters on the ML phylogeny. The first cluster was phylogenetically closer to the *16S rRNA* sequences of saprophytic (S1) and other environmental *Leptospira* species with 100% bt support, while the second cluster was found genetically distant from all *Leptospira* species but closely related to the 16S rRNA of *Leptonema illini* (bt support: 97%). Phylogenetic tree showing position of *Leptospira* classification from 16S dataset is shown in S3 File.

## Microbial composition profiling and *Leptospira* classification and identification from direct sequencing

A wide range of potentially pathogenic and water-associated microbial sequences were detected from directly sequenced soil (1,438 unique genera) and water (371 unique genera) samples. From those, 102 reads (soil: 67 reads; water: 34 reads) from 12 different *Leptospira sp.* were identified from soil and water samples. Fig 2 summarizes microbial classification profiles of direct sequencing results. Clade S1 *Leptospira spp.* reads were identified in both soil and

**Table 2. Overall read classification from all the sequencing methods used in this study** [1]**-Culture enrichment and metagenomic sequencing;** [2]**- Direct metagenomic sequencing;** [3]**-16S amplification-based sequencing.**

| Source | Total reads | Classified | Chordate | Unclassified | Microbial | Bacterial | Accession |
|--------|-------------|------------|----------|--------------|-----------|-----------|-----------|
| Water[1] | 233,994 | 69.20% | 0.01% | 30.80% | 69.20% | 69.10% | SAMN30236584 |
| Soil[1] | 237,064 | 65.60% | 0.00% | 34.40% | 65.60% | 65.50% | SAMN30236583 |
| Water[2] | 7,425 | 83.10% | 0.05% | 16.90% | 83.10% | 82.80% | SAMN30236626 |
| Soil[2] | 438,190 | 90.20% | 0.01% | 9.84% | 90.20% | 90% | SAMN30236627 |
| Water[3] | 78,756 | 99.38% | 0.00% | 0.62% | 0.00% | 99.38% | SAMN30236587 |
| Soil[3] | 68,056 | 99.66% | 0.00% | 0.34% | 0.00% | 99.66% | SAMN30236588 |

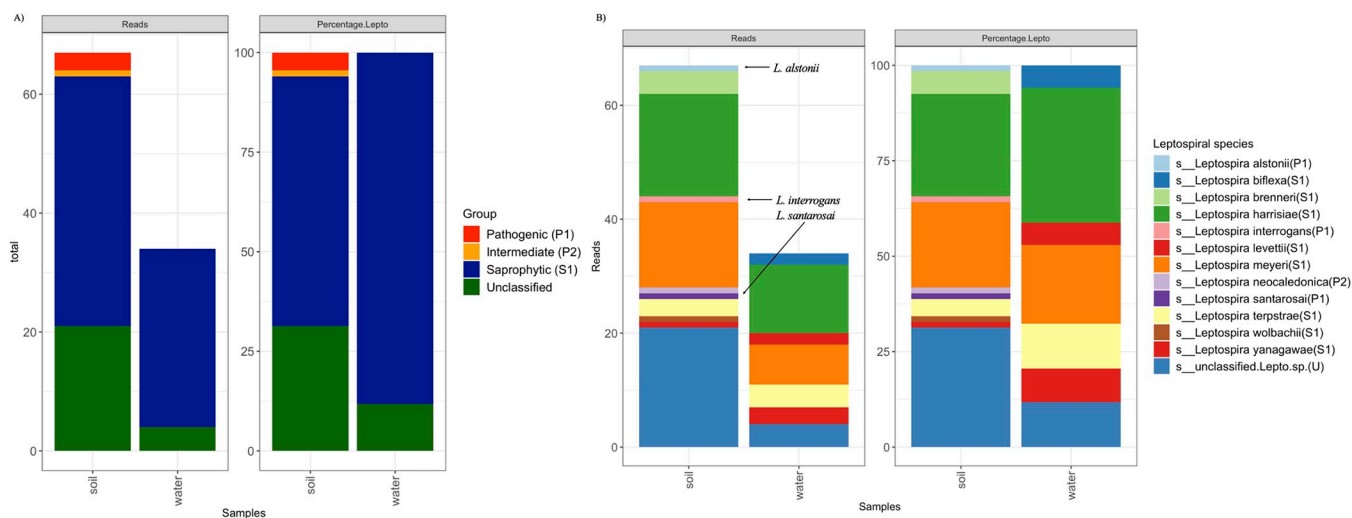

**Fig 2. *Leptospira* composition profiles for directly sequenced soil and water samples.** 2A. Proportion of *Leptospira* clades identified; 2B. *Leptospira* species-level classification. P1 clade species identified in the soil sample is labeled in the figure. The clade of each *Leptospira* species is annotated in the parenthesis behind species names in the figure legend (P1: Pathogenic; P2: Intermediate; S1: Saprophytic clade 1; U: Unclassified).

water samples. *Leptospira spp.* from saprophytic clade 2 (S2) were not identified in soil or water sample. Furthermore, pathogenic (P1) and intermediate (P2) *Leptospira spp.* reads were identified in the soil sample with low coverage (Fig 2A). Only three reads of the P1 *Leptospira sp.* (1 read from *L. interrogans;* 1 from *L. alstonii*; 1 read from *L.santarosai*) and one read of the P2 *Leptospira sp.* (*L. neocaledonica*) were identified from the soil sample (Fig 2B). In addition, around 27% and 12% of *Leptospira* reads identified in the soil and water samples could not be classified at the species level.

## Microbial profiling of the enrichment culture

We pooled positive culture samples from water and soil, prepared a composite sample for each, and proceeded with sequencing. For samples sequenced with culture enrichment, 1,325 unique microbial genera were identified across all samples (Fig 3A), with over 60% of all reads

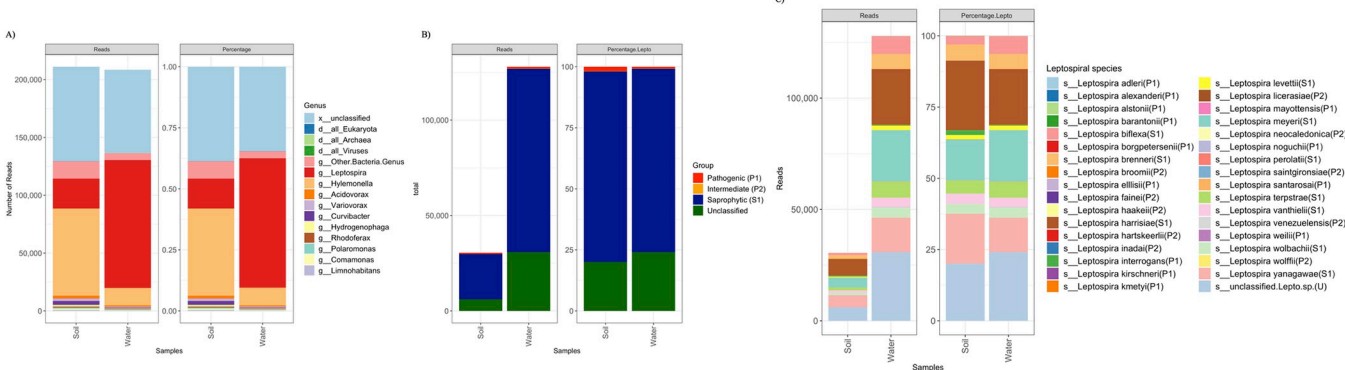

**Fig 3. Microbial classification summary statistics of pooled enrichment cultures of soil and water samples.** A) The overall microbial compositions classified in the culture enriched soil and water samples. B) The number and percentage of different clades of *Leptospira* reads identified in the soil and water samples. C) The number and percentage of reads classified under different *Leptospira* species in enrichment cultures sequenced are presented. Clade of each *Leptospira* species is annotated in the parenthesis behind species names (P1: Pathogenic; P2: Intermediate; S1: Saprophytic 1; U: Unclassified). 3A: General microbial profile, 3B: Proportion of *Leptospira* clades identified, 3C: *Leptospira* species-level classification.

classified as bacteria (Table 2). In total, 34 unique *Leptospira spp*. were identified in the enrichment cultures. It is interesting to note that 98% and 99% of *Leptospira* reads in the soil and water samples were classified under either a clade S1 species or an unclassified species (Fig 3B). Eleven clade P1 species in the soil from 551 (1.8%) reads and 13 clade P1 species in the water from 859 (0.7%) total reads were identified (Fig 3C). In addition, 9 clade P2 species in soil from 33 reads (0.1%) and 10 clade P2 species in water from 141 reads (0.1%) were identified (Fig 3C). The two additional P1 species identified in water but not soil sample are *L.borgpetersenii* and *L.elllisii*. Fig 3 summarizes microbial classification profiles of enrichment and sequencing results.

## Discussion

There is a critical knowledge gap on various aspects of environmental presence, survival, and persistence of *Leptospira*. Exposure to contaminated soil and water is a major risk factor for acquiring leptospirosis in humans and animals. The maintenance of bacteria in the soil and water and its dispersal during extreme weather events may increase the number of cases during such events. Therefore, this work was focused on evaluating and improving *Leptospira* detection from environmental samples. In this study, we observed variations in *Leptospira* detection when different techniques were applied to water and soil samples.

The original *Leptospira* taxonomy divided this genus into two species, pathogenic *L. interrogans* and saprophytic *L. biflexa* based on phenotypic characteristics [1]. These two species consist of numerous serovars based on their serologic reactivity. Later DNA hybridization studies revealed species that were classified into pathogenic, intermediate, and saprophytic species. Whole-genome sequencing projects further characterized *Leptospira* genomes to subclades including P1 (pathogenic), P2 (intermediately pathogenic), S1 (saprophytic subclade 1), and S2 (saprophytic subclade 2) revealing many genetic attributes that correlate with virulence and pathogenicity [21–23]. S1 *Leptospira* sequences were confirmed in both water and soil samples in larger proportions but no S2 *Leptospira* sequences were observed in any of our samples. The presence of the P1 and P2 groups was primarily observed in the soil. A recent systematic review also supported the presence of *Leptospira* in soil and its potential dispersion during extreme events of soil disturbance [24–26]. The bacteria may utilize the environmental conditions in the damp soil to undergo low-level proliferation enabling their persistence in the soil and subsequent transmission to susceptible hosts. Hence reservoir animal kidneys may not be the only source for *Leptospira* transmission and maintenance.

Amplification of *16s rRNA* and sequencing is a very common method used for microbial profiling of environmental samples, however, our data shows that *16s rRNA*-based metagenomics may not detect the low-level presence of *Leptospira* in environmental samples. The technique of enrichment culture followed by metagenomic sequencing improved the detection of a diverse set of pathogenic and nonpathogenic *Leptospira* in the soil and water samples at a single site. Our findings agree with many recent investigations on environmental samples identifying increased diversity of *Leptospira* species in environmental samples [24–26]. These findings emphasize the need to explore the environment as a potential reservoir of pathogenic *Leptospira*.

Analyzing environmental samples can be challenging due to the abundance and diversity of organisms present. To study a specific group of organisms in a sample, such as *Leptospira*, methods such as filtration, amplification, and selective culturing can be implemented to remove other environmental organisms that may out-compete and prevent identification of the target bacteria.

A variety of methods are used to detect *Leptospira* from environmental samples. PCR is a widely used method, and multiple gene targets have been evaluated [27]. We used three

different types of PCR with variable outcomes. PCR directly from soil or water samples did not confirm the presence of pathogenic *Leptospira*. Growing *Leptospira* in culture can be challenging. Our enrichment culture procedure evaluated various media and antimicrobial supplements following sequential filtration and sedimentation for the recovery and detection of *Leptospira*. In previous studies, filtration methods were utilized to accomplish different goals. Some studies used filters that had a large pore size of 0.7 μm to remove or capture bacteria and other studies used filters that were 0.2 μm to capture *Leptospira* on the filter [11,28]. Our methodology used a sequential filtration process to improve the efficacy. First, we used a 40 μm filter to catch large debris that could block the smaller filter and impede filtration. Then the resulting filtrate was allowed to pass through a 0.45 μm filter to remove larger bacteria, assuming that *Leptospira* with a width of 0.1 μm would pass through the filter. This double filtration method aimed to concentrate *Leptospira* in the samples, increase the chance of recovery, and reduce contamination.

Unlike the direct PCR, culture enrichment followed by PCR could detect *Leptospira* DNA in these samples. *Leptospira* culture in the presence of selective antimicrobial inhibitors might have allowed the replication of *Leptospira* while inhibiting major contaminants. Culture enrichment followed by sequencing allowed a better understanding of the diversity of *Leptospira* species present in these samples. Therefore, the sequential application of traditional and molecular methods will improve the pathogen detection and characterization from environmental samples.

Out of the PCR targets, we used the *16s* primers to amplify DNA from pathogenic and non-pathogenic *Leptospira*, *lipL32* primers to amplify DNA from pathogenic *Leptospira*. In our study, the direct PCR screening only detected an extremely low amount of saprophytic *Leptospira* DNA from the soil sample and none from the water. Intrinsic differences in amplification efficiencies and the level of original target sequences present in the samples might be a factor that contributed to the lack of detection by our direct PCR methods.

An experimental study on *Leptospira* survival in soil and water microcosms suggested the inability of *Leptospira* to multiply in environmental sites and the environment [29]. Interestingly, a recent study experimentally evaluated the suitability of water-logged soil as a medium for *Leptospira* growth [30]. They concluded that *Leptospira* can remain in the soil for longer periods in a resting state and proliferate when they come into contact with water. In bodies of water where the soil has not been recently disturbed, pathogenic *Leptospira* may be present but at DNA levels not detectable by direct real-time PCR. The limit of detection in many studies are based on spiked samples; however, heterogeneity of environmental samples may affect the sensitivity of detection. A detection limit of $10^1$ to $10^2$ *Leptospira*/mL of blood is suggested, but a higher level of *Leptospira* may be required for environmental samples due to a higher level of PCR inhibition, competing bacteria, and DNA degradation from environmental contaminates that might be present in these samples [26,31,32]. It is worth noting that direct PCR from environmental samples does not validate the presence of viable *Leptospira*. In contrast, enrichment culture followed by PCR or sequencing allows the confirmation of viable bacteria and is potentially a better method for assessing environmental maintenance and transmission risk.

Implementing *16s rRNA* amplification allows bacterial DNA in samples to be selectively amplified. This is especially useful in environmental samples, since there can be contamination.

Recently, a cost-effective workflow for microbiological profiling using targeted Nanopore sequencing of freshwater detected the presence of *Leptospira* [11]. We also used a similar method described in this study, and *16s rRNA* was amplified using barcoded custom primers followed by Nanopore sequencing. Surprisingly, only a few reads from *Leptospira* spp. were detected by these methods. This could be attributed to the nature of pathogenic *Leptospira* and

its propensity to maintain at low levels in the environment or larger more abundant environmental DNA crowding out pathogenic *Leptospira* strains limiting the number of reads obtained during sequencing. In our study, the direct sequencing from the samples resulted in a greater number of reads compared to *16s rRNA* sequencing workflow and allowed better detection of pathogenic *Leptospira*.

Next-generation sequencing allows us to study the abundance and diversity of microbial populations in environmental samples. Long-read sequencing has allowed us to study the complete genomes of organisms that are not culturable or found in association with other organisms in the environment [33,34]. The DNA of bacteria, protozoa, and animals could all be sequenced from one sample to help investigate the microbiome found in soil, water, and biological fluids [33,34]. Previously pure, cultured samples had to be used to analyze the genome of an organism, but new metagenomic sequencing technology allows complex contaminated samples to be analyzed. Commercial platforms such as Illumina and Ion Torrent are widely used for this purpose based on short-read sequencing technology. The ONT Nanopore and PacBio sequencing systems use long-read sequencing methods, and ONT nanopore method offered us a cost-effective and user-friendly platform without the need for robust equipment. One of the limitations of this system is that the pores can become clogged and create a physical barrier. Larger and more prevalent DNA will pass through the pores and possibly clog the pore before the less prevalent genomes can be sequenced subsequently resulting in lower sequence output.

Based on our findings, we propose that enrichment culture followed by real-time PCR be used for *Leptospira* surveillance of the environment. Enrichment culture followed by sequencing can be used to unravel the diversity of *Leptospira* species present in these samples. Our future studies will attempt to evaluate optimal sample volume, incubation time, and cost-effectiveness for routine environmental surveillance procedures for the detection and characterization of *Leptospira sp*. We also anticipate on isolating mixed cultures of *Leptospira* obtained in this study to purity and further characterize the pathogenic species obtained in this study.

## Supporting information

**S1 File. Growth of *Leptospira* like organisms in various cultures.** The bar charts grouped here represent the levels (0–4) of *Leptospira* growth in water(A) and soil (B) cultures over a period of four weeks. Each bar chart displays the growth for one of the four medias used along with the different selective antimicrobials added to some cultures. (A: EMJH Media, B: Fletcher, C: Korthof, D: Stuart)
(DOCX)

**S2 File.** A representative result graph of PCR on enrichment cultures: Bar charts display the Cq values of soil (A) and water (B) cultures tested after 4 weeks of incubation. Sample names along the x axis are types of media used to culture samples and the selective antimicrobial treatment. Cultures were tested with real-time PCR using *lipL32* and *16sRNA* gene markers.
(DOCX)

**S3 File. Phylogenetic tree showing position of *Leptospira* classification from 16S dataset.** Highlighted in blue and pointed to by a red arrow are *Leptospira* reads collected from soil (A) and water (B) samples in this study. Number labelled next to each node is the bootstrap support value for this specific node. Please note that *L. macculloughii* was a misidentified species from the mixed culture of *L. meyeri* and *L. levetti (24)*. Therefore, in the Figs 2 and 3, Reads classified as *L. macculloughii* was grouped together with the unclassified *Leptospira* spp.
(DOCX)

## Author Contributions

**Conceptualization:** Sreekumari Rajeev.

**Formal analysis:** Myranda Gorman, Ruijie Xu, Liliana C. M. Salvador, Sreekumari Rajeev.

**Funding acquisition:** Sreekumari Rajeev.

**Investigation:** Myranda Gorman, Dhani Prakoso, Sreekumari Rajeev.

**Methodology:** Myranda Gorman, Ruijie Xu, Dhani Prakoso, Liliana C. M. Salvador, Sreekumari Rajeev.

**Project administration:** Sreekumari Rajeev.

**Resources:** Sreekumari Rajeev.

**Supervision:** Dhani Prakoso, Liliana C. M. Salvador, Sreekumari Rajeev.

**Validation:** Myranda Gorman, Sreekumari Rajeev.

**Visualization:** Myranda Gorman, Ruijie Xu, Sreekumari Rajeev.

**Writing – original draft:** Myranda Gorman, Ruijie Xu, Dhani Prakoso, Liliana C. M. Salvador, Sreekumari Rajeev.

**Writing – review & editing:** Myranda Gorman, Ruijie Xu, Dhani Prakoso, Liliana C. M. Salvador, Sreekumari Rajeev.

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
