## [Decision Letter · Decision Letter 0]

19 Jul 2022

Dear Dr. Rajeev,

Thank you very much for submitting your manuscript "Leptospira enrichment culture followed by ONT Nanopore sequencing allows better detection of Leptospira presence and diversity in water and soil samples." for consideration at PLOS Neglected Tropical Diseases. As with all papers reviewed by the journal, your manuscript was reviewed by members of the editorial board and by several independent reviewers. In light of the reviews (below this email), we would like to invite the resubmission of a significantly-revised version that takes into account the reviewers' comments. 

All reviewers acknowledged the importance of the data presented in this manuscript and seem to believe it will be a valuable contribution to the field. However, as pointed out by two of the reviewers, the manuscript requires major revisions to improve clarity and data presentation. Please, individually address all the concerns raised by the reviewers before submitting an updated version of your work.

We cannot make any decision about publication until we have seen the revised manuscript and your response to the reviewers' comments. Your revised manuscript is also likely to be sent to reviewers for further evaluation.

Sincerely,

Andre Alex Grassmann, PhD

Guest Editor

Justin Remais

Section Editor

All reviewers acknowledged the importance of the data presented in this manuscript and seem to believe it will be a valuable contribution to the field. However, as pointed out by two of the reviewers, the manuscript requires major revisions to improve clarity and data presentation. Please, individually address all the concerns raised by the reviewers before submitting an updated version of your work.

Reviewer's Responses to Questions

**Key Review Criteria Required for Acceptance?**

**Methods**

-Are the objectives of the study clearly articulated with a clear testable hypothesis stated?

-Is the study design appropriate to address the stated objectives?

-Is the population clearly described and appropriate for the hypothesis being tested?

-Is the sample size sufficient to ensure adequate power to address the hypothesis being tested?

-Were correct statistical analysis used to support conclusions?

-Are there concerns about ethical or regulatory requirements being met?

Reviewer #1: The objectives of the paper are clear and the methods and experimental design are appropriate.

Reviewer #2: The present paper uses a set of experiments for better detection of Leptospira in water and soil samples. I have a few concerns,

1. It would be good to have a picture of the study site. Further detailing of the study site will be helpful.

2. Line 101: It is mentioned that 50g of the soil sample was collected. But in the Fig 1 and this line, it says 25g. Please clarify.

3. Line 112: I don't understand the rationale for adding 10^7 Leptospira/ml in the control sample. 

4. Please add the primer sequences in a separate table.

5. Why was the cut off set at 40? This is extremely high for analysis. Please clarify.

6. Line 274: Mention full form of NJ.

7 The Fig 3 description is coming before figure 2 in the text. Please make sure all the figures numbers and description come in succession. If needed, re format the whole manuscript so that this is followed. It made the manuscript very difficult to read.

8. This above point is also applicable for all the Tables. Please re format the manuscript.

9. Line 163: Was MAT tried for the samples? Please clarify why it wasn't used for Leptospira detection.

Reviewer #3: Some methodological parts are not clear.

- Lines 89-92 Although the amount of water/soil seems to be quite sufficient for downstream processing, it is not clear whether both types of samples were taken from a single point. In that case, sampling from several spots may (or may not) disclose more diversity in the Leptospira species detected.

- Lines 98-99 “We spiked one of the 100 mL aliquots with Leptospira interrogans serovar Copenhageni (107 bacteria per mL) to use as the control, and the second aliquot was designated as the test sample”. It is not clear whether the specified concentration is that of the initial inoculum, or the final concentration in the 100 mL aliquot.

- Lines 99-100 “The samples were further divided into 50 mL aliquots for PCR, sequencing, and culturing. A 100 mL sample was divided into three 50 mL aliquots”. How is that? I assume that one tube was used for PCR and sequencing, while the other was used for culture, it may be more specific.

- Lines 101-102 “For the processing of soil samples, the 25 g of soil was divided between two flasks and then mixed with 100 mL of phosphate-buffered saline (PBS)”. It was previously stated that 50 g were taken. So, was only half used or were 25g/25g processed in parallel?

- Lines 107-108 “We spiked one of the aliquots with Leptospira interrogans serovar Copenhageni (107 bacteria per-mL) and designated it as "control"”. Again, it is not clear whether this was the concentration of the initial culture or the concentration in the 75 mL. 

- Lines 175-176 “A composite of positive samples of culture and soil was used to reduce the cost of testing”. It is unclear if the positive cultures from water and soil samples were mixed, or the positive culture from the water alone, and the positive soil samples separately.

**Results**

-Does the analysis presented match the analysis plan?

-Are the results clearly and completely presented?

-Are the figures (Tables, Images) of sufficient quality for clarity?

Reviewer #1: Yes, and analysis and results are coherent.

Reviewer #2: 1. Fig 1 relating to the methods is very simple. Please re draw it in a better way with more details for clarity.

2. Fig 2 has 16 and 38 on top of the first bars. What does this mean? Please clarify.

3. What is the implication for not getting lipl32 in the soil test and water test? Please explain. Why do the authors think that this may be the case?

4. Figure 3 and 4 are of very bad quality. Please use high resolution images.

5. The phylogenetic trees in the supporting information should have bootstrapping values on the axes.

6. Why do the authors think the trend of Leptospira growth was different for the different media as shown in S2?

Reviewer #3: Direct PCR results from environmental samples

- Lines 234-236 Although the authors acknowledge that the results for 23S are controversial, as they are detected in the tested soil, and the spiked water, but not in the spiked soil and the tested water, it is not explained why this could happen. I fear an exchange between tested soil and tested water samples. That would make more sense. On the other hand, I assume the variation in Cq is due to technical replicates in a single qPCR run, is that correct? Did the authors perform the qPCR more than once, in independent experiments?

- As for Figure 2 also, considering they apparently used the same spike-in method for both samples, and the exponential relationship between DNA concentration and Cq values, the spiked water and the spiked soil samples showed considerable variation in terms of lipL32 detection, coming mostly from L. interrogans added as control. The spike-in method needs to be standardized and/or the final concentration of bacteria in each spiked sample needs to be better clarified (see comments above).

Culture results

- Lines 250-251 “Overall, EMJH cultures with 5-FU or STAFF were favorable for Leptospira growth for the water test group. For the soil test samples, the culture results were more variable”. Yes, but with some differences. EMJH + 5-FU reaches the higher amount of Leptospira-like microorganisms in the second week, while EMJH + STAFF in the third week. After the second week, the bacterial load in EMJH + 5-FU decreases. No bar graphs are shown for the soil samples. Was the bacterial growth evaluated more than once or is it a single tube measurement? From which combination, EMJH + 5-FU or EMJH + STAFF was the DNA extracted and then sequenced?

Microbial composition profiling and Leptospira classification from 16S dataset

- Lines 270-271 “A very low number of reads were classified under the phylum taxon "Spirochaetota" in the water (1 read) and soil (9 reads) samples when 16S rRNA gene sequence dataset was analyzed”. In Supplementary Figure 3, however, panel A showing the phylogenetic tree of the water samples highlights 9 branches, and panel B (soil samples), a single branch. Contrary to what is indicated in the text.

Microbial profiling of the enrichment culture

- Lines 309-310 “It is interesting to note that 98% and 99% of leptospiral reads in the soil and water samples were classified under either a saprophytic species or an unclassified species”. I do not think this is surprising, as the enrichment was done over a 4-week period, and saprophytic Leptospira has a shorter doubling time than pathogenic Leptospira (approximately 8h versus 20h). It would be interesting to take samples for sequencing at different time points (one per week, for example, or after 2 and 4 weeks) and compare the distribution between pathogenic and saprophytic leptospires over time.

- It would have been interesting to compare to low "activity" sample that could serve as a negative control.

- The sample have been pooled, what would have been the results without pooling them?. Is this representative or is there only one sample that account for all the results?.

**Conclusions**

-Are the conclusions supported by the data presented?

-Are the limitations of analysis clearly described?

-Do the authors discuss how these data can be helpful to advance our understanding of the topic under study?

-Is public health relevance addressed?

Reviewer #1: Yes.

Reviewer #2: 1. The discussion section doesn't adequately address the significance of the results obtained.

2. The public health relevance hasn't been addressed well.

Reviewer #3: The discussion is clearly too long as compared to the rest of the document and tend to over-interpret the data. 

- Lines 370-374 “Our methodology used a double filtration system to improve the efficacy. First, we used a 40 μm filter to catch large debris that could block the smaller filter and impede filtration. Then the resulting filtrate was allowed to pass through a 0.45 μm filter to remove larger bacteria, assuming that Leptospira with a width of 0.1 μm would pass through the filter”. Double filtration is not described in methodology.

- Lines 376-378 “Unlike the direct PCR, culture enrichment followed by PCR could detect Leptospira DNA in these samples”. I find it relevant to show the Cq graphs for PCR on enriched cultures.

**Editorial and Data Presentation Modifications?**

Reviewer #1: Accept

Reviewer #2: The figures are of very low quality.

Reviewer #3: - It is known that 16S sometimes produces ambiguous results in terms of species classification, and this has been mentioned in several articles focusing on Leptospira classification. In this regard, it would have been good to try alternative targets as well. In a recent publication, Wilkinson and collaborators (2021) used glmU to detect specific species in culture and culture-independent samples, also using Nanopore. Although restricted to a few species, they obtained good results.

- The classification as pathogenic and intermediates is becoming obsolete. It may be pertinent to update to the more recent classification (P1, P2, S1, S2) in some parts of the document.

- Line 314-315 “Figure 4 summarizes microbial classification profiles of direct sequencing results”. It should refer to “enrichment and sequencing” instead of “direct sequencing results”.

- The “Accession” column in Table 2 is empty.

- the supplementary figure S1 and Figure 1 could be better designed to include the different samples and replicates as compared to just repeating the text. I am not sure are the replicates have been used.

**Summary and General Comments**

Reviewer #1: The authors present a comparative analysis of different methods to detect Leptospira from environmental samples, including use of 16S amplicons and whole-genome using the Oxford Nanopore Platform. Considering that environmental survailance is an important applications for molecular methods, including NGS, and the relevance of Leptospirosis in public health, specially in tropical and under-development countries, the paper provides a very important contribution to the field.

Reviewer #2: The authors present an interesting study but the manuscript is not well edited and formatted. The Fig numbers are not in succession and the English needs to be proof read by a native English speaker. Also, the discussion section doesn't adequately address the significance of the results obtained. The manuscript needs major work to address these concerns.

Reviewer #3: In the present study, Gorman and colleagues apply different approaches to identify Leptospira spp. in environmental samples, i.e., soil and water. The methods assessed include PCR, direct metagenomic sequencing, and culture with subsequent PCR or metagenomic sequencing. The results obtained highlight the advantages of using culture enrichment and then PCR/sequencing, as opposed to direct PCR or 16S-based metagenomics. While the authors underline as a novelty the fact that enriched cultures are more suitable for detecting abundance and diversity in environmental samples, this is not surprising. Leptospira is slower growing than other bacteria present in the environment, so an enrichment would certainly be helpful for its detection in a complex sample. However, the impact of the article resides in the comparison of the most commonly used techniques such as direct PCR, or 16S-based metagenomics of samples, versus traditional enrichment methods along with ONT Nanopore sequencing, in detecting Leptospira spp. They even test several culture media and antimicrobial agents to ensure Leptospira growth. Being aware of the advantages of Nanopore sequencing (short-term, cost-effective sample processing), this sounds promising as an epidemiological surveillance tool. 

Please not that the sequencing data was not availlable.

PLOS authors have the option to publish the peer review history of their article (what does this mean?). If published, this will include your full peer review and any attached files.

Reviewer #1: Yes: Frederico Schmitt Kremer

Reviewer #2: No

Reviewer #3: No
---

## [Decision Letter · Decision Letter 1]

27 Sep 2022

Dear Dr. Rajeev,

Thank you very much for submitting your manuscript "Leptospira enrichment culture followed by ONT Nanopore sequencing allows better detection of Leptospira presence and diversity in water and soil samples." for consideration at PLOS Neglected Tropical Diseases. As with all papers reviewed by the journal, your manuscript was reviewed by members of the editorial board and by several independent reviewers. The reviewers appreciated the attention to an important topic. Based on the reviews, we are likely to accept this manuscript for publication, providing that you modify the manuscript according to the review recommendations. 

In particular, please, consider the comments and suggestions raised by Reviewer 3 before submitting a revised version of your manuscript.

Sincerely,

Andre Alex Grassmann, PhD

Guest Editor

Justin Remais

Section Editor

The reviewers appreciated your efforts to improve the manuscript, which will be suitable for publication after some minor adjustments. Please, consider the comments and suggestions raised by Reviewer 3 before submitting an updated version of your manuscript.

Reviewer's Responses to Questions

**Key Review Criteria Required for Acceptance?**

**Methods**

-Are the objectives of the study clearly articulated with a clear testable hypothesis stated?

-Is the study design appropriate to address the stated objectives?

-Is the population clearly described and appropriate for the hypothesis being tested?

-Is the sample size sufficient to ensure adequate power to address the hypothesis being tested?

-Were correct statistical analysis used to support conclusions?

-Are there concerns about ethical or regulatory requirements being met?

Reviewer #2: Yes

Reviewer #3: (No Response)

**Results**

-Does the analysis presented match the analysis plan?

-Are the results clearly and completely presented?

-Are the figures (Tables, Images) of sufficient quality for clarity?

Reviewer #2: Yes

Reviewer #3: (No Response)

**Conclusions**

-Are the conclusions supported by the data presented?

-Are the limitations of analysis clearly described?

-Do the authors discuss how these data can be helpful to advance our understanding of the topic under study?

-Is public health relevance addressed?

Reviewer #2: Yes

Reviewer #3: (No Response)

**Editorial and Data Presentation Modifications?**

Reviewer #2: Yes

Reviewer #3: (No Response)

**Summary and General Comments**

Reviewer #2: The authors have satisfactorily answered or edited the manuscript based on the recommendations. I would like to thank the authors for revising the manuscript.

Reviewer #3: This is a re-submission of the article by Gorman et al., in which the authors discuss the advantages of combined enrichment culture and Nanopore sequencing in the detection of Leptospira spp. from environmental samples, as compared to more commonly used techniques (direct PCR, 16S-based metagenomics).

General comments:

- One of the strongest criticisms I had in the first version was the lack of clarity in several methodological parts. The methodology is now much more detailed, not just what was specifically asked, but throughout the entire section. Several comments/suggestions were also properly addressed. It could have been ok to have figures without copyrights representing the protocol. 

- The 23S-based PCR is still a bit problematic. Lines 391-392 specifically states that the primers used amplify non-pathogenic Leptospira. However, 23S is detected in the “water spiked” sample inoculated with pathogenic L. interrogans, but not in the “water test” sample. I wondered if “water test” and “soil test” had been interchanged, which is also supported by the fact that 23S in both “soil test” and “water spiked” have comparable Cq (considering the associated error). I understand that the authors had used controls during sample processing and ruled out such an exchange. So, if there was not a DNA swap when doing PCR, the 23S-based PCR is not valid and not worthy to be shown. 

- Figure 2 and 3 lists L. macculloughii among the detected species. Although L. macculloughii was described in Thibeaux 2018, it was later not considered again (Vincent et al., 2019), since it was determined to be the result of a mixed culture between L. meyeri and L. levetti.

Minor comments/suggestions:

- Lines 284-285. Differences in font type.

- Lines 313 and 354. Leptospira should be in italics.

- Lines 315-316. It is mentioned that two additional P1 species were found in water samples compared to soil samples. From Figure 3C it is not so evident which were these two species, so it might be worth mentioning them in the text.

- Line 316. “Figure 4C” is cited but does not exist in this version, so it should be changed to “Figure 3C”. Also, the caption in Figure 3 contains cross-references to Figure 4 from the previous version.

- Line 366. Punctuation marks: two consecutive full stops at the end of a sentence.

PLOS authors have the option to publish the peer review history of their article (what does this mean?). If published, this will include your full peer review and any attached files.

Reviewer #2: No

Reviewer #3: No

Figure Files:

Data Requirements:

Reproducibility:

References

---

## [Editor Report · Decision Letter 2]

10 Oct 2022

Dear Dr. Rajeev,

We are pleased to inform you that your manuscript 'Leptospira enrichment culture followed by ONT Nanopore sequencing allows better detection of Leptospira presence and diversity in water and soil samples.' has been provisionally accepted for publication in PLOS Neglected Tropical Diseases.

Best regards,

Andre Alex Grassmann, PhD

Guest Editor

Justin Remais

Section Editor

---

## [Editor Report · Acceptance letter]

21 Oct 2022

Dear Dr. Rajeev,

We are delighted to inform you that your manuscript, "Leptospira enrichment culture followed by ONT metagenomic sequencing allows better detection of Leptospira presence and diversity in water and soil samples.," has been formally accepted for publication in PLOS Neglected Tropical Diseases.

Best regards,

Shaden Kamhawi

co-Editor-in-Chief

Paul Brindley

co-Editor-in-Chief
